# The Influence of TBC Aging on Crack Propagation Due to Foreign Object Impact

**DOI:** 10.3390/ma12091488

**Published:** 2019-05-08

**Authors:** Przemysław Golewski, Tomasz Sadowski

**Affiliations:** Faculty of Civil Engineering and Architecture, Lublin University of Technology, Nadbystrzycka 38, 20-618 Lublin, Poland; p.golewski@pollub.pl

**Keywords:** thermal barrier coating system, FOD, impact, FEM, aging

## Abstract

While a plane is maneuvering before take-off and landing, some solid particles (e.g. sand, dust, soil) may get into the engine with air. A vast majority of them are stopped by the compressor blades, but the smaller ones can get into a hot part of the engine and cause erosion. A pneumatic laboratory work station was built in order to investigate the impact of foreign object damage (FOD) particles with a diameter of 4 mm. Cylindrical samples with a diameter of 30 mm were used, each having a thermal barrier coating (TBC) deposited by the air plasma spray (APS) method with the application of yttria-stabilized zirconia (YSZ). Sample aging was performed for four ranges: 48, 89, 185, and 353 h at the temperature of 1000 °C. After aging, samples were subjected to impacts made with different energies. Various damage images were captured depending on the aging time and impact velocity. Numerical studies led to the determination of how the incidence angle of a foreign object and the blade temperature affected the number of elements that became damaged during impact. It was found that impacts perpendicular to the surface were the most dangerous, while heating the blade to the operating temperature resulted in a 27% decrease in the number of elements damaged during impact when compared to the cold blade.

## 1. Introduction

Thermal barrier coatings (TBCs) have been used for many years in order to protect critical turbine engine parts (i.e., blades, combustion chambers) from high temperatures and aggressive environments. Along with the development of computer-aided design/computer-aided engineering (CAD/CAE) programs and the enhancement in the computational power of workstations, finite element method (FEM) simulations [1,2,3,4,5,6,7,8,9,10,11,12,13,14,15,16,17] are also conducted in addition to laboratory experiments. These simulations are used to analyze complex thermo-mechanical states [1,4,6], heat flow for a variety of coating materials [3], the effect of cooling channels on cooling efficiency [2], or TBC damage during impact loads [8].

In [18], the authors investigated the erosion resistance of five TBC systems: a TBC deposited by plasma spray–physical vapor deposition (PS–PVD), one segmented air plasma sprayed (APS) system, one highly segmented APS, one porous APS, and one TBC system deposited by electron beam-physical vapor deposition (EB-PVD). The worst results were obtained for the porous APS. The lowest erosion was obtained for the hybrid PS–PVD system. The authors analyzed the influence of two impact angles (30° and 90°) and two impact speeds (40 and 104 m/s), finding that different damage effects might occur during foreign object damage (FOD) in the TBC system. These different effects were reduced to three models for the EB-PVD system, as shown in [19]. Subsequently, the authors performed FEM simulations in two dimensions to define the stress impact of a foreign object on a column structure. An analytical dependence regarding the maximum penetration and generated stresses during impact was also determined. In [20], the authors gave a mathematical function describing impact depth, depending on parameters such as the impact force, foreign object diameter, and yield stresses. A schematic design of a laboratory station for FOD investigation was given in [21], consisting of three elements: a holder that enables sample heating, a device for measuring speed, and a gas gun. The sample was heated to 1100 °C and the FOD aluminum particles had a velocity ranging from 50 to 250 m/s. Simultaneously, 2D numerical simulations were conducted. The problem of the aging effect was undertaken in [22]. A relationship between the impact angle of FOD particles and the erosion speed was determined. Aging was conducted at 1100 and 1500 °C for 24, 30, and 100 h. The mass loss of the samples aged for 24 h at 1500 °C was four times higher than that observed for the samples without aging. Two types of FOD for the EB-PVD system were also distinguished. In [23], a distinction was made between erosion, corrosion, and corrosion–erosion. The last one is associated with calcium–magnesium–alumina–silicates (CMASs) and is a separate and very complex problem.

All previous studies refer to flat samples with a TBC system. However, real objects such as blades have complex geometric shapes. In [24], the authors compared three methods: a 3D simulation of the FOD impact on the leading edge of the blade, a 2D photoelastic method, and an analytical method. The results of the stress concentration were similar for all methods. However, in their study, the authors did not take account of the TBC system. The paper by Carter [25] is very general and covers issues such as mechanical damage, damage due to high temperature, creep, and corrosion. Load types should also be taken into account when TBC systems are analyzed. Kumar and Balasubramanian [26] give an overview of issues such as sintering, residual stresses, oxidation, corrosion, high temperature corrosion, erosion, CMAS, and damage mechanisms. To prevent erosion, various techniques may also be used, such as the sintering of the outer layer [27] or the deposition of the aluminum layer [28]. In [27], the “laser glazing” method was used, consisting of a laser melting the TBC outer layer and thereby increasing its density. Erosion tests were also performed on standard samples. The same results were only achieved for the impact angle of 90°, and for the angles of 30° and 60° the glazing effect was found to be very favorable. However, this method is both labor- and energy-intensive and would be difficult to apply to large objects. A different TBC system involving the deposition of an Al layer with a thickness of 15 μm was proposed in [28]. The samples were subjected to aging in a vacuum for 1 h at 700 °C and for 5 h at 900 °C. Phase transformations and the in situ synthesis of ZrO_2_ and Al occur on the interface of Al and TC (top-coat) at high temperatures. The product is an Al_3_Zr compound with high toughness, a high melting point, and chemical stability. The laboratory test stand was a self-modified device made from a sand-blasting machine. Al_2_O_3_ particles of 100 μm dimension were used, and the erosion angle was set equal to 90° with a distance of 20 cm. The samples without Al coating showed a mass loss amounting to 4 × 10^−4^ g/cm^2^, while for the samples with Al coating, the mass loss was about 2.6 × 10^−4^ g/cm^2^. It can therefore be stated that this method helps improve TBC properties; nevertheless, like the previous method, it is time- and energy- consuming and cannot be used in practice for large objects.

Summing up, current works focus on examining the erosion caused by small particles of several tens to several hundreds of μm. However, there are no studies investigating the erosion caused by large objects that may be located at the airstrip. Considering FEM simulations, the focus is mainly put on 2D models. Given the above, this work aimed to experimentally investigate the impact of FOD particles with a diameter of 4 mm and link the experimental findings with the results of an FEM simulation analyzing the FOD impact on the turbine blade with a TBC system for the cold blade and the blade heated to the working temperature alike. In effect, the above-mentioned knowledge gap could be filled to some extent. Conducting numerical research for objects with real geometries is very important from the point of view of turbine engine design. 

## 2. Experimental Research

Experimental research was conducted on samples with 30 mm diameter with a TBC system. The substrate thickness was 2 mm, the bond-coat (BC) thickness was 100 μm, and the thickness of a ceramic coating (TC–top-coat) was 250 μm. The samples were first subjected to aging in a muffle furnace at 1000 °C for 48, 89, 185, and 353 h. After that, indents were made in the samples using the test station described in [8]. A steel ball with a diameter of 4 mm was used as an indenter. The samples were fixed in an interference-preventing wooden enclosure, which was then screwed to a cross table. The cross table ensured the precise distribution of indents with an accuracy of 0.01 mm. A total of 30 impacts were made, as shown in Figure 1. 

The impact speed was measured indirectly by measuring pressure in the pressure vessel. Knowing the characteristics of the work station shown in [8], it was possible to determine the impact speed values, listed in Table 1.

After making indents, two cross-section cuts were made in each sample by the section method. Then, the samples were subjected to grinding and polishing. Afterwards, the samples were macroscopically examined with a Quanta FEG 250 (Thermo Fisher Scientific, Hillsboro, OR, USA) scanning electron microscope.

Figure 1 shows the effect of heating on the erosion caused by the steel ball indenter. A common feature of the analyzed samples is that there was no delamination between the substrate and the bond coat, and there was only plastic deformation. Another characteristic of the presented indents is that, in addition to the total erosion of the ceramics in the region of impact, delamination also occurred on the TC/BC interface (Figure 2). On the other hand, differences were observed with respect to the amount of ceramic material that became completely eroded. As for the samples that were not subjected to aging, the diameter of the removed part was about 0.8 mm, and its shape was close to spherical.

For aging times from 48 to 185 h, the ceramic edges in the area of erosion were irregular, and their diameters were 0.76, 1, and 1.05 mm, respectively. The largest diameter of the removed ceramic fragment was observed in the sample subjected to aging for 353 h, and it was approximately 2.06 mm. 

In one of the images for the first indent (0 h), we can clearly distinguish different damage areas of the ceramic top-coat:

Area I—The complete removal of the TC,

Area II—The presence of numerous horizontal cracks. This area was very narrow near the BC interface and expanded with increasing TC height, 

Area III—The only visible damage was delamination at the TC/BC interface.

These three areas are shown in Figure 3a at a magnification of 500×, and area II is shown at a magnification of 1000× in Figure 3b. 

Upon further observation of the impact area at 400× magnification, three patterns of crack formation could be distinguished depending on the aging time. These three patterns concerned the crack path in relation to the BC/TC interface at which the thermally-grown oxide (TGO) layer grew due to high temperature and the presence of oxygen (Figure 4). 

In Pattern I (Figure 4a), the crack extended along the interface even when it was folded. With increasing aging time, diffusion took place and the BC/TC interface became more durable. In Pattern II (Figure 4b), the crack passed tangentially to the tops of interface folds, and the ceramic layer remained in the valleys. Pattern III could be distinguished (Figure 4c) at the maximum aging time, where the crack occurred only in the TC layer, tens of micrometers away from interface fold tops. 

The crack formation patterns above should be considered in FEM analysis. This may prove to be troublesome because FEM simulation requires 3D modelling of the TBC structure at the micro level, which is more difficult to perform because of the large size of the blades and combustion chambers, which in turn results in a large number of finite elements. 

Figure 5 shows the effect of kinetic energy for the aging time of 89 h. A common feature of all indents is the occurrence of delamination at the BC/TC interface. Analyzing the influence of kinetic energy, one can also distinguish at least three models of damage. The first entailed the complete removal of the TC in the impact area (Indents 1,2); the second was characterized by lower energy, which means that a small-thickness ceramic material remained in the place of impact (Indent 3); in the third model (Indents 4,5,6), a significant amount of ceramic material remained in the place of impact, but there were numerous cracks. At this point, the question can be raised as to whether the TBC system could continue working under such damage and, if so, for how long (in the case of Model 3)? Also, what value will the thermal conductivity have? It should be noted that the impact area was small (approx. 0.8 mm^2^) in comparison to the blade surface area, which was several thousand square millimeters. Taking the heat flow into consideration, it can be concluded that an internal cooling system could deal with this kind of damage. This, however, is a separate problem that requires further research and numerical simulations.

## 3. Numerical Studies 

The literature on the subject refers to the 3D numerical modelling of blades in a limited way. Furthermore, all laboratory samples are flat, and their mounting methods are often not specified in the test descriptions. Note that the substrate stiffness also affects the erosion level at single impact. In order to fill this knowledge gap to some extent, an attempt was made to model a 3D blade with a TBC system subjected to impact.

In this work, a fragment of the stationary blade with a cooling channel was numerically modelled in the Abaqus program. The geometrical model consisted of four parts (Figure 6): a foreign object, a blade, and two fragments of coating (TBC 1 and TBC 2) that were made so that it was easier to dense the mesh, as shown in the figure. However, no separate geometries were used for the BC and TC. The differences in the material were taken into account by assigning sections to previously partitioned parts of the TBC.

The total number of finite elements in the whole model amounted to 200,408, including 197,350 of C3D8R elements and 3050 C3D6 elements. The number of finite elements used in individual parts of the assembly is given in Table 2.

Numerical simulations of foreign object impact were performed for two variants: cold and heated blades. In the heated blade variant, a coupled temperature–displacement step was used. The outer surface of the blade was described with the sink temperature of 800 °C and the surface film condition of 300 W/m^2^K, while the channel surface cooling was described with the sink temperature of 20 °C and the surface film condition of 150 W/m^2^K. In this variant, the steady-state process was considered to obtain the temperature distribution shown in Figure 7.

For the FOD calculations in Abaqus, the dynamic/explicit step type was selected. The simulation duration was set equal to 0.0005 s. In every tested variant, the initial speed of a foreign object (i.e., a ball with 4 mm diameter) was kept constant at 74.84 m/s. This value was used for the first indent in the experiments. Five angles of foreign object incidence were analyzed, as shown in Figure 8. All degrees of freedom were taken away from the lower part of the blade. 

Because the simulations were carried out at different temperatures, the material data also had to be dependent on temperature, as shown in Table 3. Both the bond-coat and ceramic top-coat were considered as linear elastic materials. The substrate was modelled as an elastic–plastic material with isotropic hardening but without damage, since the experiments demonstrated that damage only occurred in the ceramic top-coat layer. The substrate material was described with the yield strength σ_Y_ = 190 MPa and the tensile strength σ_T_ = 630 MPa with the elongation at break A = 0.45.

In order to describe the damage of TC material, similarly to [8], a brittle cracking material model was used. It was a smeared crack model, which means that it did not track single cracks. Crack initiation was based on a simple Rankine hypothesis that assumed a crack occurred when the maximum tensile stress reached the tensile strength $\sigma_{t}^{I}$, as in [30,31,32,33,34,35,36,37,38,39,40,41,42,43]. Once this criterion was satisfied, the elements were removed from the mesh, which affected the stress values and rigidity of the material. Therefore, a denser mesh should be used to model the real process more accurately. In this case, for the TBC 2 part, the global element size was set equal to 0.25 mm. In addition, four elements along the TC partition thickness were used. The finite element mesh was made dense enough to enable the observation of changes in the number of deleted elements, depending on the angle of incidence and the temperature (Table 4).

Analyzing Table 4, we can conclude that the most dangerous impacts were those perpendicular to the surface. For the non-heated blade, the number of damaged elements was 6–9 times greater than in the case of a foreign object striking tangentially to the surface. An interesting observation can be made regarding the blade heated to the working temperature and the orthogonal incidence. Here, the number of deleted elements was reduced by approximately 27%. This was due to material deformation which occurred after heating (Figure 9), its value being equal to 1/5 of the strains causing damage. This further compensated for the deformation caused by the impact, which reduced the number of deleted elements for certain angles, especially those normal to the surface. This phenomenon requires further study using a higher-density mesh, which also indicates the need of numerical analyses for different temperatures.

## 4. Conclusions

This paper presented the results of the dynamic impact of a foreign object in the form of a steel ball with 4 mm diameter at the TBC system. Different indent diameters were obtained depending on the aging time, ranging from 0.8 mm (0 h) to 2.06 mm (353 h). Depending on the region of crack occurrence, three areas were distinguished and described: an area with total erosion, an area with horizontal cracks, and an area with only delamination. By examining the TC/BC interface, three fracture models were distinguished and described. The microscopic observations revealed that in any event there was no damage on the substrate/bond-coat interface, and the impact speed affected the number of ceramic coatings that underwent erosion.

An attempt was made to link the results obtained for a real TBC system with the numerical findings of a simulation, taking account of finite element damage. Strain values calibrated for the model corresponding to the real sample dimensions were transferred to a 3D brittle cracking material model. In the paper, the different impact angles of a foreign object that could get into the engine were also considered. Impacts perpendicular to the surface were found to be the most dangerous; on the other hand, it was found that the element damage was reduced by about 27% when the blade was heated to the working temperature, the proposed model allows for analyzing a large number of impact combinations in certain areas, with different particle velocities, dimensions, and temperatures. In effect, the above model could become a valuable tool for engineers who design turbine engines.

## Figures and Tables

**Figure 1 materials-12-01488-f001:**
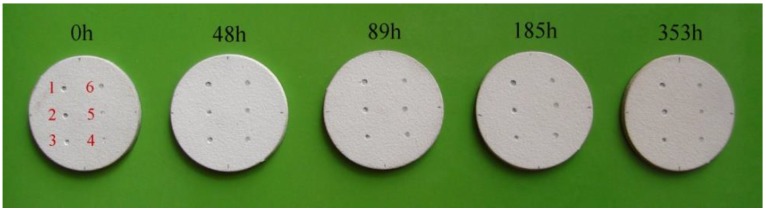
View of the samples after impact testing.

**Figure 2 materials-12-01488-f002:**
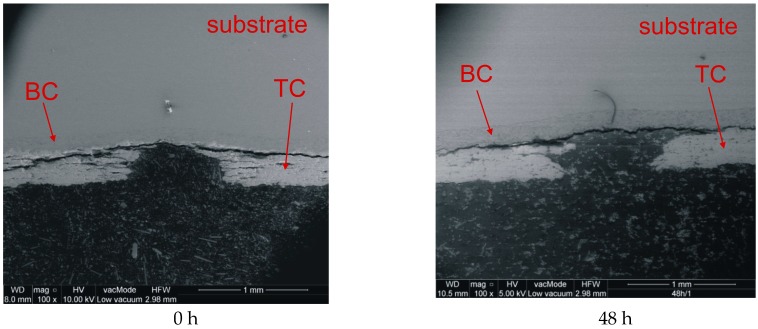
Images of indents for impact no. 1 and various aging times. BC: bond coat; TC: top coat.

**Figure 3 materials-12-01488-f003:**
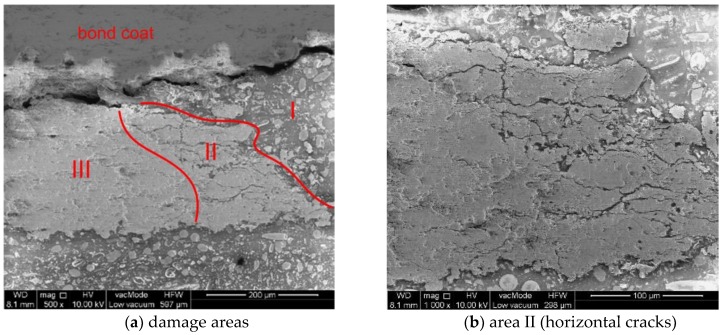
Damage areas of the top-coat.

**Figure 4 materials-12-01488-f004:**
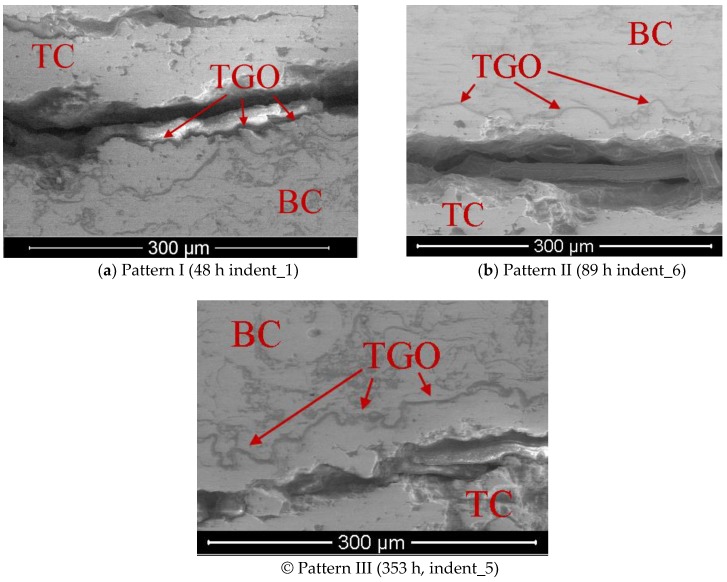
Crack formation patterns, depending on the aging time. TGO: thermally-grown oxide.

**Figure 5 materials-12-01488-f005:**
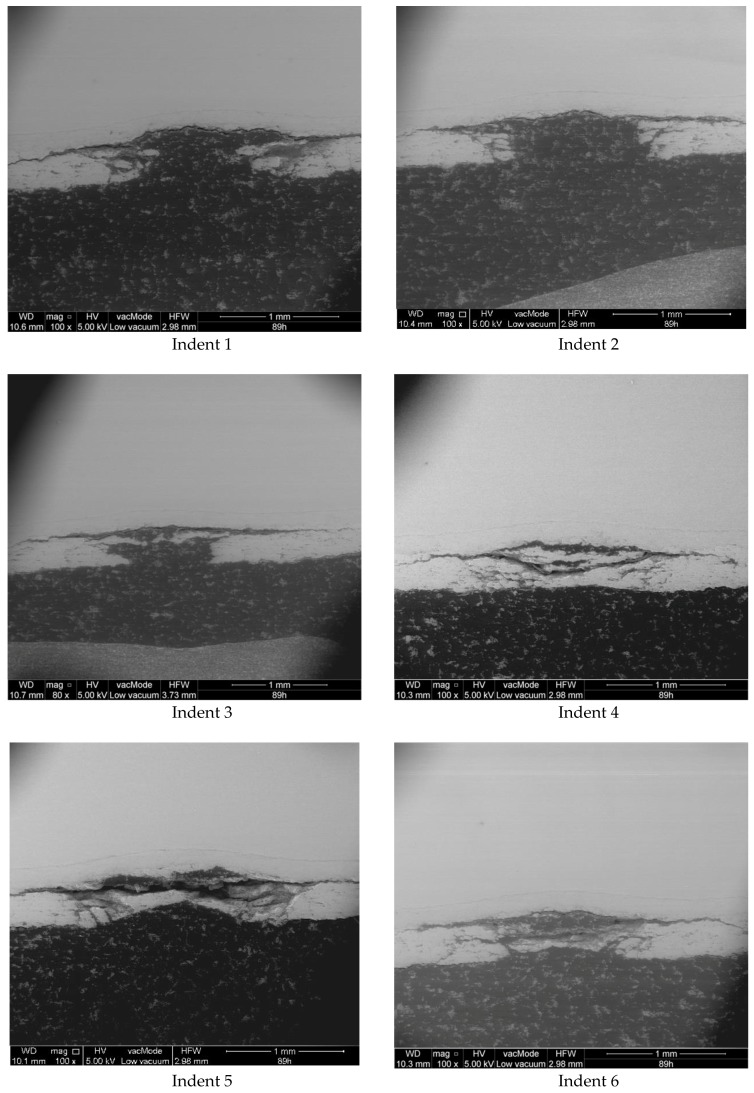
Images of indents for the aging time set to 89 h and different speeds.

**Figure 6 materials-12-01488-f006:**
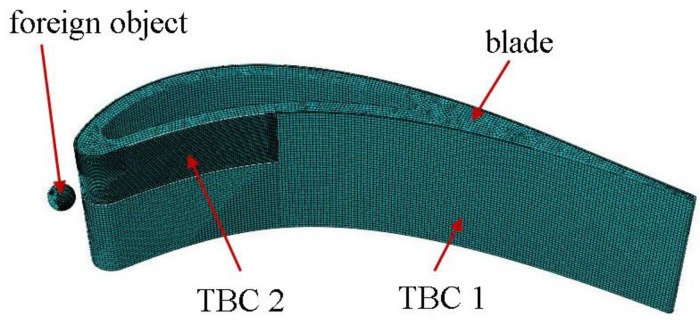
Finite element mesh. TBC: thermal barrier coating.

**Figure 7 materials-12-01488-f007:**
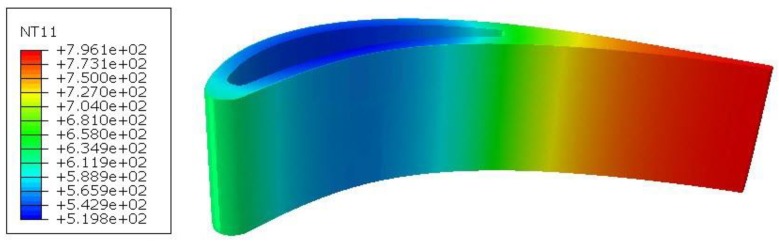
Temperature distribution in the blade.

**Figure 8 materials-12-01488-f008:**
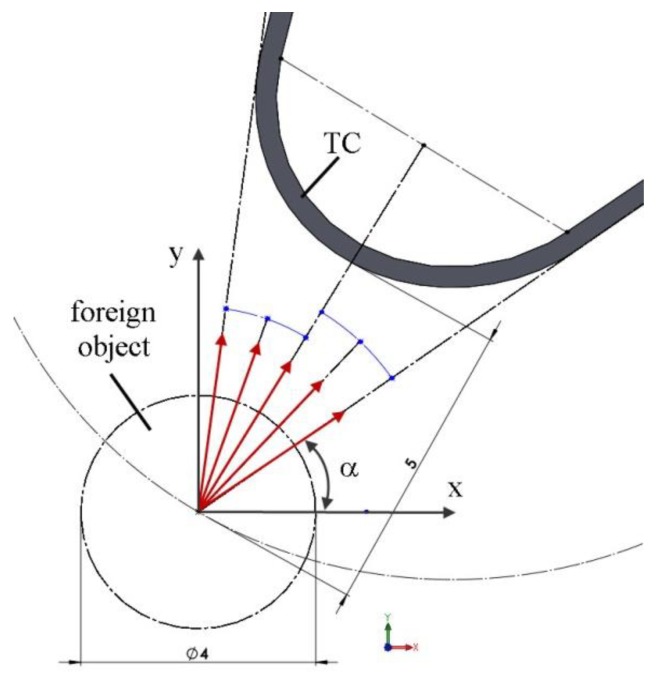
Angles of foreign object incidence.

**Figure 9 materials-12-01488-f009:**
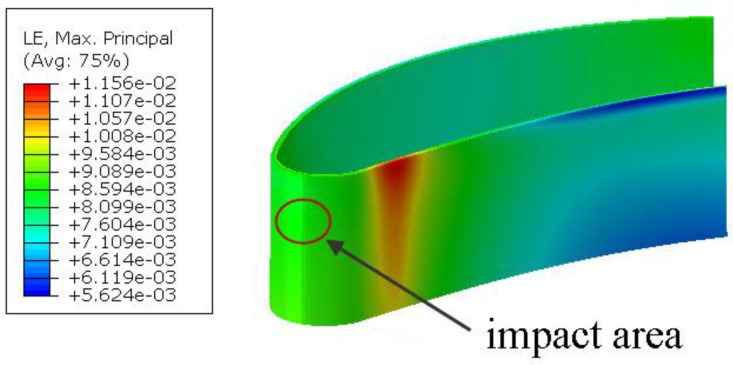
Strains caused by blade heating.

**Table 1 materials-12-01488-t001:** Impact speed (m/s).

No. of Impacted Point	0 h	48 h	89 h	185 h	353 h
1	74.84	74.84	74.84	74.84	74.73
2	74.07	73.91	74.22	74.22	74.07
3	73.39	72.81	73.39	73.39	73.20
4	72.18	71.74	71.96	72.40	71.96
5	71.04	70.31	70.80	71.04	70.56
6	69.81	69.05	69.56	70.06	69.56

**Table 2 materials-12-01488-t002:** Number of finite elements.

Type of Finite Element	Foreign Object	TBC 1	TBC 2	Blade
C3D8R	12,160	35,240	57,800	92,050
C3D6	1408	-	-	1650

**Table 3 materials-12-01488-t003:** Material data [29]. YSZ: yttria-stabilized zirconia.

Material Property	Temp.	Substrate	Bond-Coat (BC)	Top-Coat (YSZ)
Young modulus (GPa)	25 °C	200	200	85
	1000 °C	150	120	35
Poisson’s ratio	25 °C	0.33	0.3	0.1
	1000 °C	0.33	0.3	0.1
Thermal exp. coeff. (10^−6^ 1/K)	100 °C	10.8	10	9.2
	1000 °C	16.8	17.5	10.5
Density (kg/m^3^)	25 °C	8500	7380	3610
	1000 °C	8500	7030	3510
Specific heat (J/kgK)	25 °C	440	450	505
	1000 °C	700	980	630
Thermal conductivity (W/mK)	25 °C	8.9	10.8	0.9
	1000 °C	21.6	32.1	0.3

**Table 4 materials-12-01488-t004:** Number of damaged elements.

α	82.39°	70.48°	58.57°	46.76°	34.84°
Cold blade	6	8	18	8	2
Hot blade	6	8	13	10	4

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
