# Peer review of "The Influence of TBC Aging on Crack Propagation Due to Foreign Object Impact"

_materials, 2019, doi:10.3390/ma12091488_

Round 1
Reviewer 1 Report
1. In order to make the paper clear for the wide readership of the Journal, it is advisable to explain all the abbreviations (i.e., TBC, APS, YSZ, CAD/CAE, PC-PVD, EB-PVD, etc.) at the first use.
2. The closing sentence of the Introduction (“Concluding numerical research…”) looks more like an opening sentence.
3. The paper needs a serious revision for English by a native speaker. For example, the following changes are advisable:
Line 11: While a plane is maneuvering before take-off and landing,
Line 60: for all methods. However, in
Line 61, 62: The paper [16] is very general and covers such issues as: mechanical
Line 66: also be used, e.g. sintering of
Line 67: In [17], the “laser glazing”
Line 73: On the next step, samples were aging in
Line 79: the samples with Al, it was
Line 95: As an indenter, a steel ball
Line 102: work station shown in [8], it was possible
Line 106: operation, the macroscopic
Line 110-111: A common feature of the presented samples is that there occurs no delamination between the substrate and the bond coat for all cases and there is only the plastic deformation.
Line 115: subjected to aging, the diameter
Line 130: impact area at magnification of 400x, we can distinguish
Line 135: In model I (fig. 4a), the crack extends
Line 136: increase of aging time, the diffusion
Line 138: For the maximum aging time, model III can be
Line 155: first, the one consisting in
Line 156: second one, for
Line 157: the third model (indents 4, 5, 6) covers a significant
Line 162: in comparison to the blade
Line 191: foreign object, i.e. a sphere
Line 198: Due to the fact that the simulations
Line 203: substrate material, the yield strength
Line 221: Analyzing Table 4, we can
Line 243: In the paper, an attempt to link results for real
Author Response
Response to Reviewer 1
Thank you for your valuable remarks and questions which increase the manuscript quality.
Re: All remarks of the reviewer were included in revised version of the manuscript.
Serious revision for English by a native speaker was introduced using red colour.
Reviewer 2 Report
The list of detailed remarks is given below:
The aging times should be more regular (e.g. 96 h instead 89 h).
For wider description it will be better to show that not only Authors have been worked with FEM simulations of TBC systems.
In experimental part there are few questions:
what was the substrate used in experiment?
what was the material for BC and its granulation?
what was the granulation of YSZ for TBC (probably it was ZrO2 + 8 wt% Y2O3)?
Fig. 2 - the quality of the SEM images is quite poor, additionaly there is no marks of BC and TBC, as well as the delamination areas.
Fig. 3 and line 129 - Authors wrote, that in case of Fig. 3b there are 3 areas, while there is only area II (due to the caption)
There is no explanation of the mechanism of different models of cracks depending on the aging time (Fig. 4).
According to the Fig. 5 and lines 158 and 159 - if we see in details onf indent no. 4 and compare it with indent no. 5 the damage is smaller rather in case of indent no. 4 than no. 5. Please comment this phenomenon, becasue it is an exception among others runs.
Table 3 - what was the material datas source, especially in case of density? Additional, there is no information about type of substrate and BC.
Table 4 - please explain what could be an explanation of higher quantity of damages at smaller degree in case of hot blade in compare to the cold one.
Author Response
Response to Reviewer 2
Thank you for your valuable remarks and questions which increase the manuscript quality.
1. The aging times should be more regular (e. g. 96 h instead 89 h)
Re: Irregular times were due to the long cooling time of the furnace and the fact that in some cases the samples were taken out at night when there were no workers in the laboratory. In the future, we will use regular times.
2. For wider description it will be better to show, that not only Authors have been worked with FEM simulations of TBC systems.
Re: Literature review has been extended by 9 articles in the field of FEM modeling.
3. In experimental part there are few questions:
- what was the substrate used in experiment ?
Re: it was 0H18N9 stainless steel,
- what was the material for BC and its granulation ?
Re: it was NiCoCrAlY
- what was the granulation of YSZ for TBC (probably it was ZrO2 + 8 wt% Y2O3) ?
Re: yes, it was ZrO2 + 8 wt% Y2O3, but the granulation is unknown, samples were made in Polish Aviation Plants in Rzeszów.
4. Fig. 2 – the quality of the SEM images is quite poor, additionally there is no marks of BC and TBC, as well as the delamination area.
Re: In figure 2 descriptions were added: substrate, BC, TC
5. Fig. 3 and line 129 – Authors wrote, that in case of Fig 3b there are 3 areas, while there is only area II (due to the caption)
Re: The sentence in the text was changed.
6. There is no explanation of the mechanism of different models of cracks depending on the aging time (Fig. 4).
Re: The mechanism lies in the fact that additional stresses appear as a result of the increase of the TGO layer. In addition, there is a change in the chemical composition of individual layers and thus a change in mechanical properties. The oxygen content in the TC layer increased from 56.48% to 63.19 after 353 hours. Oxides share increased.
7. According to the Fig. 5 and lines 158 and 159 – if we see in details on indent no. 4 and compare it with indent no. 5 the damage is smaller rather in case of indent no. 4 than no. 5 Please comment this phenomenon, because it is an exception among others runs.
Re: This is due to the fact that the tests were performed in a small range of impact energy from 0.714J to 0.616J. The damage depends on many local factors, eg folding BC-TGO-TC layer and different thickness of individual layers. Hence, in the next studies, tests with more imprints and with larger energy intervals should be carried out.
8. Table 3 – what was the material data source, especially in case of density? Additional, there is no information about of substrate and BC.
Re: The source is the article:
Tzimas E.; Mullejans H.; Peteves S. D.; Bresser J.; Stamm W. Failure of thermal barrier coating systems under cyclic thermomechanical loading. Acta mater 2000, 48, 4699-4707.
BC material – NiCoCrAlY,
Substrate – Ni-based superalloy.
9. Table 4 – please explain what could be an explanation of higher quantity of damages at smaller degree in case of hot blades in compare to the cold one.
Re: It is difficult to answer precisely. Maybe, it is due to fact that under high temperature and impact under low angle a level of Huber – von Mises - Hencky stresses decreased and therefore we observed in the numerical example smaller numer of damaged finite elements.
Round 2
Reviewer 2 Report
In my opinion all sugestions and comments have been included by Authors.